# Impact of LEAP and CBT-AN Therapy on Improving Outcomes in Women with Anorexia Nervosa

**DOI:** 10.3390/bs13080651

**Published:** 2023-08-03

**Authors:** Phillipa Hay, Mohammed Mohsin, Liquan Liu, Stephen Touyz, Caroline Meyer, Jon Arcelus, Sloane Madden, Evelyn Attia, Kathleen M. Pike, Janet Conti

**Affiliations:** 1Mental Health Research Unit, Liverpool Hospital, South Western Sydney Local Health District, NSW Health, Liverpool, NSW 2170, Australia; 2Translational Health Research Institute, School of Medicine, Western Sydney University, Campbelltown, NSW 2560, Australia; 3Faculty of Medicine & Health, School of Clinical Medicine, University of New South Wales, Sydney, NSW 2052, Australia; 4School of Psychology, Western Sydney University, Sydney, NSW 2560, Australia; 5The MARCS Institute for Brain, Behaviour and Development, Western Sydney University, Sydney, NSW 2560, Australia; 6Inside Out Institute, University of Sydney, Sydney, NSW 2006, Australia; 7International Digital Laboratory, The University of Warwick, Coventry CV4 7AL, UK; 8Institute of Mental Health, University of Nottingham, Nottingham NG7 2TU, UK; 9Psychoneurobiology of Eating and Addictive Behaviors Group, Bellvitge Biomedical Research Institute (IDIBELL), Bellvitge University Hospital, University of Barcelona, 08908 Barcelona, Spain; 10The Sydney Children’s Hospitals Network, Westmead, Sydney, NSW 2145, Australia; 11The New York State Psychiatric Institute, Columbia University, New York, NY 10032, USA; 12World Health Organization Collaborating Centre for Capacity Building and Training in Global Mental Health, Columbia University Medical Center, New York, NY 10032, USA

**Keywords:** anorexia nervosa, eating disorder, mental health, quality of life, compulsive exercise activity therapy (LEAP)

## Abstract

Anorexia nervosa (AN) is a mental health disorder that has serious physical, emotional and social consequences. Whilst cognitive behavioural therapy for AN (CBT-AN) has demonstrated efficacy, there remains a global need to improve AN treatment. Compulsive exercise activity therapy (LEAP) is an active therapy consisting of the addition to CBT-AN of eight specific sessions that focus on exercise and motivation for behavioural change. This paper presents a secondary analysis of 74 female participants in a randomised control trial of LEAP plus CBT-AN versus CBT-AN alone. The main aim of this study was to explore putative predictors and to estimate the magnitude of changes due to LEAP for specific outcome measures. Participants (LEAP: n = 36; CBT-AN: n = 38) were assessed at three successive surveys: baseline, end of therapy, and 6 months post-therapy. The overall effect sizes for changes between baseline to end of therapy and baseline to 6-month follow-up assessment showed large effect sizes (Cohen’s *d* > = 0.80) for mental-health-related quality of life (MHRQoL), weight concern, dietary restraint, eating concern, AN stage change, and psychological distress (all *p* < 0.05). The results also indicated that several pre-treatment characteristics, including body mass index (BMI), level of eating disorder (ED) symptoms, and MHRQoL are important for identifying whether a treatment is likely to be effective. Future treatment programs should aim to optimise early improvements in BMI, ED symptoms, and MHRQoL.

## 1. Introduction

Anorexia nervosa (AN) is a serious mental health disorder characterised by body image concern and the restriction of energy intake relative to requirements, leading to a significantly low body weight [1]. AN may be associated with significant psychological comorbidity, severe medical complications, and considerable psychosocial impairment [2] and has the highest mortality rates of all mental health disorders [3]. Existing literature indicates that outcomes for individuals with AN have improved little in the second half of the 20th century [4].

A small number of studies have been completed to determine a treatment of choice for adults with AN [5]. A recent systematic review by Zeeck et al. (2018) [6] identified five randomised control trials (RCTs) with moderate to high quality that evaluated cognitive-behavioural therapy (CBT) in individualised outpatient treatment for adults with AN. Although this review found that CBT did not differ in outcome variables when compared to most other therapies (e.g., FT-AN, family based treatment for anorexia nervosa; IPT, interpersonal psychotherapy; SSCM, specialist supportive clinical management; CRT, cognitive remediation therapy; CAT, cognitive-analytic therapy), it did find CBT to have significant efficacy in improving global AN rating [7], body mass index (BMI; kg/m^2^), eating disorder (ED) quality of life (QoL), depression, social adjustment, and ED psychopathology. Furthermore, CBT is accepted as a leading therapy by professionals and in clinical guidelines [8]. Despite the demonstrated efficacy of CBT, the associated poor outcomes, high mortality rate, and low recovery rate of AN highlight the importance of ongoing research into effective treatment and prevention strategies [5].

Compulsive exercise is an important feature of AN and has received little coverage in clinical research [9]. Compulsive exercise can be defined as an individual’s extremely driven and inflexible exercise patterns which they have a perceived lack of capacity to stop despite awareness of their adverse effects [10]. Studies have reported that increased compulsive exercise has been significantly associated with higher levels of ED symptoms [11], weight and shape concern, dietary restraint [12], body dissatisfaction, and drive for thinness [13]. Some studies have reported that compulsive exercise is associated with poorer treatment outcomes, including an earlier time of relapse and poor long-term outcomes [14,15]. Due to the relationship between compulsive exercise, AN psychopathology, and outcomes, specific strategies are required to address compulsive exercise [16].

A CBT for compulsive exercise in EDs, namely compulsive exercise activity therapy (LEAP)—also known as the Loughborough eating disorders activity programme (LEAP)—has been developed in order to improve treatment outcomes by reducing pathological exercise cognition and subsequent compulsive exercise behaviours [10]. LEAP is an active therapy, with responsibility for behaviour change residing with the patient as consistent with motivational interviewing approaches [17]. LEAP is based on the cognitive behavioural theory of the maintenance of compulsive exercise, which extends the trans-diagnostic theory of EDs developed by Fairburn [18]. The extension was added after significant evidence to suggest a reciprocally reinforcing relationship between compulsive exercise and eating pathology. Alongside the essential mechanism of shape and weight concern, the extended model identifies three key factors that may maintain compulsive exercise, including dysfunctional affect regulation, compulsivity, and perfectionism and rigidity [19]. LEAP and a modification for young people in hospital (“Junior LEAP”) has been tested and found efficacious in two RCTs [10,19], and a third is underway [20].

The study, “Taking a LEAP Forward in the Treatment of Anorexia Nervosa: A Randomised Trial” [19] aimed to enhance the effectiveness of current treatments by including LEAP into a pre-existing manualised CBT for AN (CBT-AN). The study found that both CBT-AN and CBT-AN with LEAP resulted in improved attitudes and beliefs toward exercise and general improvements in ED psychopathology and BMI [19]. This study suggested that further investigation of the LEAP and CBT-AN trial is needed, specifically to identify the association of factors involving ED psychopathology with BMI, the eating disorder examination questionnaire (EDE-Q) global, the Short-Form 12-item Mental Health Component Summary Scale (SF12MCS); and health-related quality of life [19]. However, in this first report, the predictors of treatment outcome were not explored and there was no in-depth examination of factors of the magnitude of change for specific outcomes over the course of therapy and a follow-up [19].

This is a secondary detailed analysis of the original data from an RCT of CBT-AN and LEAP that were previously published as a brief paper [19]. The purpose of this study was to extend others’ findings and examine factors associated with outcomes in an RCT of CBT-AN and LEAP prior to therapy (baseline), at the end of therapy, and at 6-month follow-up after therapy. Whilst specific ED symptoms, such as eating concern [21] and shape concern [22], have found to be predict poorer treatment outcomes [23,24], variability in the definition and measurement of ED features such as BMI, ED psychopathology, and EDQoL and the focus of the literature on pre-treatment baseline predictors [25] has resulted in inconsistent findings and limited interpretations of previous research. Thus, the main aim of this study was to explore putative predictors of outcome (namely, BMI, ED symptoms, readiness to change, psychological distress, EDQoL, and general health related quality of life) and to estimate the magnitude of their changes between baseline to end of therapy; baseline to 6-month follow-up; and end of therapy to 6-month follow-up. Finally, the association of specific outcome measures (BMI, EDE-Q global score, EDQoL, and SF12MCS) with participant characteristics was also examined.

## 2. Materials and Methods

### 2.1. Study Design and Participants

The current study is part of a 3-site 2-armed parallel RCT that took place in Sydney, Australia; Leicester, United Kingdom (UK); and New York, United States of America (USA) during the period of 2010–2016. The trial was registered with number ACTRN12610000585022. Participants were recruited from ED clinics and community advertising. To be eligible, participants had to be 18 years or older, retrospectively met *Diagnostic and Statistical Manual of Mental Disorders Fifth Revision* (DSM-5) criteria for anorexia nervosa (AN), have a BMI of 14 to 18.5, and have reported at least one exercise activity during the previous month. Exclusion criteria included a diagnosis of psychosis or bipolar disorder, current substance dependence, medical instability, and high risk of suicide. A total of 574 cases were assessed for eligibility, of which 78 eligible participants (UK: 39, Australia: 25, USA: 14) completed self-report assessments at baseline (496 excluded: 296 did not meet the inclusion criteria, 56 had incomplete information for assessment, and 144 declined to participate in the study). Baseline participants were assigned randomly using an external computer-generated system and stratification by location, illness subtype, and medication use into two groups—38 in LEAP and 40 in CBT-AN group—to receive therapy. Participants were blind to the treatment group, provided written informed consent, and were debriefed at the end-of-therapy session. The LEAP module utilised CBT techniques to address unhelpful exercise beliefs and behaviours. The CBT-AN group completed 34 individual sessions of manualised CBT for AN over 8–10 months. Manualised CBT-AN is an active therapy aimed at restoring weight and normal eating behaviours by challenging thoughts and beliefs through behaviour change and cognitive restructuring [26]. The LEAP group completed 26 sessions of CBT-AN and eight sessions of LEAP treatment. Therapy was twice weekly for 4 weeks and weekly thereafter. LEAP treatment involved equipping participants with skills and knowledge to aid them in regaining control of their exercise behaviour to partake in appropriate levels of exercise [10]. Core modalities include cognitive restructuring and behavioural experiments. Therapists were trained in both LEAP and CBT-AN, provided both therapies, and conducted two supervised pilot patient 34-session treatments each.

Baseline participants were re-assessed via self-report questionnaire at the end of therapy and 6 months after the therapy. At the end of the therapy, a total of 50 baseline participants were re-assessed (LEAP: 24, CBT-AN: 26) with a retention rate of 64.1% (50 out of 78), and at 6-month follow-up, 43 participants were assessed (LEAP: 18; CBT-AN: 25), with a retention rate of 55.1% (43 out of 78). It is to be noted that among the 78 baseline participants, only 4 were male (5.1%), and 74 were female (94.9%). The disproportionate number of males may introduce gender bias in the estimated results, and the results may not be generalizable to the overall population level. Considering gender bias in the sample, only 74 female participants (n = 74; LEAP: 36, CBT-AN: 38) are included in the final analytic sample (Appendix A).

### 2.2. Ethics

Ethics was approved for this study at each site: the Western Sydney University Human Research Ethics Committee; the Institutional Review Board at Columbia University in New York, USA; and the National Health Service Research Ethics Committee in the UK, as part of the Health Research Authority.

### 2.3. Survey Measures

#### 2.3.1. Socio-Demographic Characteristics and Body Mass Index (BMI)

Socio-demographic data were obtained at baseline, which included questions on age, marital status, highest level of education, and employment status. BMI (kg/m^2^) was recorded by objective measurement using calibrated scales and a stadiometer.

#### 2.3.2. Mental Health History

Information was obtained at baseline through questions regarding date of diagnosis and psychotropic medication use.

#### 2.3.3. Mental Health Outcome Measure

We applied the same set of mental health measures at all three assessment periods of data collection. Measures used in this paper included body mass index (kg/m^2^; BMI); the Eating Disorder Examination–Questionnaire (EDE-Q); ED-health-related quality of life (EDQoL); general physical and mental health with the SF-12 item Health Status Questionnaire; the Compulsive Exercise Test (CET); psychological distress with the Kessler-10 (K-10); and the Anorexia Nervosa Stages of Change Questionnaire (ANSOCQ).

##### Eating Disorder Examination–Questionnaire (EDE-Q)

The EDE-Q is a 28-item self-report questionnaire based on the longer clinician administered EDE interview [18]. The EDE-Q has a global score and four subscales, with higher scores indicating greater psychopathology. The four subscales assess the range and severity of diagnostic features for EDE-Q global score, which includes dietary restraint, eating concern, shape concern, and weight concern. The EDE-Q has robust psychometric properties [27]. In our study sample, Cronbach’s alpha (α) for the item pool of EDE-Q global score was 0.92 at baseline, 0.89 at the end of therapy, and 0.92 at six-month follow-up assessment.

##### Compulsive Exercise Test (CET) and Exercise Beliefs Questionnaire

The CET measures core maintaining factors of compulsive exercise for individuals with EDs [10]. It has 5 subscales, which examine avoidance and rule-driven behaviour, weight control, lack of exercise enjoyment, mood improvement, and exercise rigidity. The self-report measure has 24 items and is scored on a 6-point Likert scale. Higher scores indicate more compulsive exercise. A sample item is “I enjoy exercising”. The CET has established psychometric properties [28] and Cronbach’s α in this study sample was 0.93 at baseline, 0.95 at the end of therapy, and 0.94 at six-month follow-up assessment.

The 21-item Exercise Beliefs Questionnaire (EBQ) [29] measures of maladaptive beliefs about exercise. It has a total score and four subscales: “social desirability”—beliefs about the social consequences of not exercising (e.g., feeling inferior or judged negatively by others if not exercising); “physical appearance”—beliefs of physical unattractiveness (e.g., “look bad”) if not exercising; “mental and emotional functioning” (e.g., inability to “cope” if not exercising); and “vulnerability to disease and aging” if not exercising. The EBQ has established psychometric properties. In this study, the Cronbach α for the EBQ total was 0.95.

##### Psychological Distress (Kessler-10)

The Kessler-10 (K-10) scale is a measure of psychological distress and assesses symptoms of anxiety and depression [30]. The self-report measure has 10 items and is scored on a five-point Likert scale (1 = none, 2 = a little of the time, 3 = some of the time, 4 = most of the time, 5 = all the time), with high scores indicating greater psychological distress. A sample item is “Over the past four weeks (28 days), about how often did you feel hopeless?”. The K-10 has robust psychometrics [31] and in this study demonstrated excellent internal consistency at baseline (α = 0.92), end of therapy (α = 0.94), and six-month follow-up assessment (α = 0.92).

##### Anorexia Nervosa (AN) Stages of Change Questionnaire

The AN stage of change questionnaire (ANSOCQ) measures motivation to change in patients with AN [32]. The self-reported measure has 20 items with 5 answers, each representing the different stages of change (pre-contemplation, contemplation, preparation, action, and maintenance). Higher scores indicate a greater motivation to change. An example pre-contemplation item which assesses attitude to weight loss is “I would prefer to lose more weight”. The ANSOCQ has been found to have sound psychometrics [32] and demonstrated excellent internal consistency in this study (α = 0.91), at end of therapy (α = 0.94), and at six-month follow-up assessment (α = 0.92).

##### Health Related Quality of Life (HRQoL)

Health-related quality of life (HRQoL) was assessed with the SF-12 item Health Status Questionnaire v2 [33] and the EDQoL [34] measure. The SF-12 examines functional limitations to physical and mental health to assess quality of life. This study utilised the two subscales: the Physical Health Component Summary Scale (SF12PCS) and the Mental Health Component Summary Scale (SF12MCS), with high scores indicating better HRQoL.

The EDQoL is an eating-disorder-specific 25-item measure that examines the extent to which the participant views their AN as affecting their quality of life, with higher scores indicating lower HRQoL. The EDQoL has previously been used to assess disease-specific quality of life in patients with eating disorders [35] and patients with chronic AN. It has robust psychometric properties, and Cronbach’s α for the overall scale was 0.92 at baseline, 0.94 at the end of therapy, and 0.92 at six-month follow-up assessment.

### 2.4. Statistical Analyses

In the first step, we presented descriptive statistics of baseline sociodemographic characteristics (age, educational attainment, employment status, and marital status) and psychotropic medication for the LEAP and CBT-AN groups. Continuous variables are presented as means and standard deviation (SD), and categorical variables were presented as percentages. In the next step, we presented mean total scores with 95% confidence interval (CIs) for all outcome measures (namely, body mass index, EDE-Q global score, weight concern, dietary restraint, eating concern, shape concern, compulsive exercise test, exercise beliefs questionnaire, ANSOCQ, SF12MCS, SF12PCS, EDQoL, psychological distress) by LEAP and CBT-AN participants for those who completed assessments at three points of time: baseline (n = 74; LEAP: 36, CBT-AN: 38), end of therapy (n = 48; LEAP: 22, CBT-AN: 26), and 6-month follow-up (n = 41; LEAP: 17, CBT-AN: 24). The overall mean total scores for all outcome measures were compared between LEAP and CBT-AN groups across all three assessment periods. As the outcome measures are continuous, we applied mixed models to examine the adjusted interaction effects for time (three assessment periods: baseline, end of therapy, and six-month follow-up) and group (LEAP, CBT-AN) effects. The method allows flexibility in modelling covariance structures involving longitudinal and repeated data of a correlated type, considering within-subject, time-dependent correlations. In addition to this multiple group comparison test between “baseline (T0) vs. end of therapy (T1) assessment”; baseline vs. 6-month follow-up (T2) assessment; and T1 vs. T2 were conducted through the statistical technique ANOVA (analysis of variance) for repeated measures. The Bonferroni statistical test was applied to examine the pairwise significant differences across time points (that is, T0 vs. T1; T0 vs. T2; and T1 vs. T2) for all outcome measures controlling for LEAP and CBT-AN participants. In order to reduce the risk of type-1 error, which may occur in multiple comparison testing for multiple endpoints, the levels of significance i.e., all *p*-values < 0.05 from the Bonferroni statistical test were adjusted. Based on the two-way comparisons, we computed effect sizes (Cohen’s *d*) [36] for each outcome as an indication of the magnitude of change in treatment outcomes from baseline to end of therapy (T0 to T1), baseline to 6-month follow-up (T0 to T2), and from end of therapy to 6-month follow-up (T1 to T2). We applied the established thresholds for interpreting the effect sizes, with a Cohen’s *d* of 0.2 denoting a small effect, 0.5 a medium effect, 0.8 and above a large effect [36]. We only included those who participated in assessments at T0, T1, and T2, respectively.

Next, we examined the association of main outcome measures (BMI, EDE-Q global score, SF12MCS, EDQoL) with patient characteristics and all other outcome measures. For categorical variables, the mean total scores are presented by subgroups; and for continuous variables, correlation coefficients are presented. It is to be noted that the study was part of a 3-site, 2-armed parallel randomised control trial that took place in Sydney, Australia (n = 23); Leicester, United Kingdom (n = 38), and New York, United States of America (n = 13). In order to examine the differences of measures by country of trial controlling for two randomisation groups require larger sample size. As the initial analyses shows that the main outcome measures at baseline for BMI (*p* = 0.10), EDE-Q global score (*p* = 0.133), and EDQoL (*p* = 0.59) do not differ significantly by study site (country of trial), to obtain stable estimates, this study used combined participants (n = 74) of three trial places as the analytic sample for statistical analysis.

Measures found to be statistically significant with each of the four main outcome measures (BMI, EDE-Q global score, SF12MCS, EDQoL) are included in multiple linear regression analysis to further examine the relative contribution of each of these predictive measures adjusting for age and treatment group. Considering the three assessment periods (T0, T1, T2), three multiple regression analyses were carried out for each of the main outcome measures, i.e., a total of 12 multiple regression analysis were performed. We used SPSS 27 [37] to conduct all the statistical analysis.

## 3. Results

### 3.1. Participants’ Characteristics

The sociodemographic and clinical characteristics of study participants at baseline assessment are reported in Table 1. At baseline assessment, there were no significant differences for any of the socio-demographic characteristics between two treatment arms LEAP (n = 36) and CBT-AN (n = 38) participants. The mean age of the whole sample of 74 female participants was 27.1 years (*SD* = 8.7). Almost half (50.0%) of all participants had completed or were participating in university-level education, 41.9% completed up to high-school-level education, and a minority (8.1%) had obtained a trade certificate. More than two thirds (68.9%) of the participants were single, 28.4% were married, and over half (56.6%) were employed. At the baseline survey, 54.1% reported having purging behaviour, and a similar proportion (54.1%) had taken psychotropic medication (Table 1). Participants had an average BMI of 16.5 (*SD* = 1.1).

### 3.2. Outcome Measures across Three Assessment Periods

Table 2 reports the results for the outcome measures based on the mixed models with interaction effects for time (assessment periods: baseline, end of therapy, and six-month follow-up) and treatment group (LEAP, CB-TN). The results also revealed that all outcomes differed across assessment periods within each group. Except for the compulsive exercise test (CET, *p* = 0.028), all other outcome measures (body mass index, EDE-Q global score, weight concern, dietary restraint, eating concern, shape concern, exercise beliefs questionnaire, ANSOCQ, SF12MCS, SF12PCS, EDQoL, K-10) showed no significant differences by group, adjusting for time effects over the three assessment periods. In addition, none of the outcomes showed statistically significant interactions between groups in terms of assessment time.

Table 3 reports the mean scores with 95% CIs for all outcome measures by LEAP and CBT-AN participants those completed assessment at baseline (T0; n = 74), end of therapy (T1; n = 48), and at 6-month follow-up after therapy (T2; n = 41) respectively. At baseline assessment, there were no significant differences in mean scores for any of the outcome measures (BMI, EDE-Q global score, weight concern, dietary restraint, eating concern, shape concern, compulsive exercise test, ANSOCQ, EDQoL, SF12MCS, SF12PCS, psychological distress) between LEAP (n = 36) and CBT-AN (n = 38) groups. At the end of therapy, the mean scores for dietary restraint (*p* = 0.020) and *CET* (*p* = 0.006) were found to be significantly higher among the CBT-AN group as compared to the LEAP group. At the 6-month follow-up, except for the CET score (*p* = 0.016), all other measures do not differ significantly between LEAP and CBT-AN participants.

The mean outcome measures score for both LEAP and CBT-AN participants at T1 and T2 showed significant reductions as compared to T0 for EDE-Q global score, weight concern, dietary restraint, eating concern, shape concern, EDQoL, and psychological distress and a significant increase in ANSOCQ and psychological distress score (all Ps < 0.05) (Table 3). Irrespective of LEAP and CBT-AN participants, for scores of exercise beliefs questionnaire and SF12PCS, there were no significant differences observed across three assessment periods (Table 3).

Appendix B reports the pairwise significant mean differences (raw and adjusted *p*-values) across different time points (i.e., T0 vs. T1; T0 vs. T2; and T1 vs. T2) through the Bonferroni statistical test controlling for the LEAP and CBT-AN group participants. Each matched sample comprised all participants who completed each outcome measure between the two time points under comparison. The results in Appendix B showed a similar pattern as presented in Table 3. Both LEAP and CBT-AN participants reported significantly lower scores on all mental health indices between T0 and T1 and T0 and T2 (all *p* < 0.05). It is to be noted that between T0 and T1 and T0 and T2, based on raw (unadjusted) *p*-values, 85% of findings were significant at *p* < 0.01, and after applying corrections of *p*-values, 81% of findings become significant at *p* < 0.01. As shown on the table in Appendix B, the mean BMI score for both groups increased from T0 to T1 and was found to be statistically significant among CBT-AN participants only (*p* < 0.001) but was not found to be statistically significant among LEAP participants. The mean score of exercise beliefs between T0 and T1 for participants of both intervention groups had overall *p* = 0.001 but for CBT-AN participants was *p* = 0.03 (adjusted *p* = 0.09). Furthermore, the mean scores of all the mental health measures significantly differed between T0 and T2 except for exercise beliefs within each intervention group and SF12MCS among LEAP participants, but combined results for participants in both intervention groups reached significance (*p* = 0.015 for exercise beliefs and *p* = 0.001 for SF12MCS). Between T1 and T2, there were no significant differences for any of the outcome measures except for dietary restraint among the CBT-AN group (*p* =0.016, adjusted *p* = 0.048). This might be because of shorter follow-up time duration.

Table 4 reports effect sizes (Cohen’s *d*) for each outcome measure as an indication of the magnitude for each outcome by LEAP and CBT-AN group participants for baseline (T0) assessment compared to end of therapy (T1), T0 compared to 6-month follow-up (T2), and T1 compared to T2 assessment. The overall effect sizes for baseline to end of therapy; and baseline to 6-month follow-up assessment showed larger effect sizes (Cohen’s d > = 0.80) for EDE-Q global score, EDQoL, SF12MCS, weight concern, dietary restraint, eating concern, ANSOCQ and psychological distress; and medium effect sizes (value of d ranges 0.50 to 0.79) for shape concern and the CET. The overall effect sizes for all measures between end of therapy to 6-months follow-up was quite small (d < 0.50), this might be because of shorter follow-up time duration. The effect sizes for baseline to end of therapy assessment showed that effect sizes for LEAP were slightly larger than the CBT-AN group for most of the measures, including EDE-Q global score (LEAP, d = −1.55; 95%CI: −2.17, −0.94 versus CBT, d = −1.02, 95%CI: −1.50, 0.53), EDQoL (LEAP, d = −1.21; 95%CI: −1.79, −0.63 versus CBT, d = −0.88; 95%CI: −1.33, −0.41), weight concern (LEAP, d = −1.08; 95%CI: −1.62, −0.51 versus CBT, d = −0.80; 95%CI: −1.40, −0.32), dietary restraint (LEAP, d = −1.14; 95%CI: −1.70, −0.56 versus CBT, d = −0.94; 95%CI: −1.42, −0.32), and eating concern (LEAP, d = −0.84; 95%CI: −1.35, −0.32 versus CBT, d = −0.82; 95%CI: −1.27, −0.32); marginally larger effect sizes for CBT were observed for ANSOCQ (LEAP, d = 1.01 versus CBT, d = 1.05) and psychological distress (LEAP, d = −0.83 versus CBT, d = −1.07). There appears to be no effect for LEAP on BMI at the end of therapy, while CBT-AN shows moderate effect size (LEAP, d = 0.04 versus CBT, d = 0.73). The effect sizes for main outcomes by patient characteristics revealed that irrespective of patient characteristics larger effect sizes observed between T0 and T2; effect sizes slightly differ by patient characteristics and treatment group and were not found to be statistically significant (Table 5). The effect of therapy was observed for both LEAP and CBT-AN group even by 10 weeks of therapy (Appendix C). The change in mean scores for the main outcome measures EDE-Q global score, EDQoL, and SF12MCS shows significant differences between baseline and 10 weeks of therapy (Appendix C).

#### 3.2.1. Association of Main Outcome Measures with Participants’ Characteristics and Measures

Findings presented in Table 6 revealed that BMI, EDE-Q global score, and EDQoL scores at baseline are not significantly associated with participant’s age, level of education, employment, and psychotropic medication status. The SF12MCS score at baseline was found to be significantly higher for employed (*p* = 0.004) and those who reported with no psychotropic medication (*p* = 0.008).

The correlation co-efficient between main outcome measures with other outcome measures and participants characteristics are presented in Table 7. Although BMI at T0 was not found to be significantly associated with any of the T0 measures, BMI at T1 was found to be significantly associated with T1 SF12MCS (r = 0.30; *p* < 0.05), EDQoL (r = −0.48, *p* < 0.01), and ANSOCQ (r = 0.34, *p* < 0.05). BMI at T2 was found to be significantly associated with T2 EDQoL (r = −0.52; *p* < 0.01), dietary restraint (r = −0.39, *p* < 0.05), eating concern (r = −0.45, *p* < 0.05) and ANSOCQ (r = 0.35, *p* < 0.05).

Results in Table 7 also revealed that EDE-Q global score at T0 was found to be significantly associated with participant’s age (r = −0.26; *p* < 0.05), T0 SF12MCS (r = −0.49; *p* < 0.01), EDQoL (r = 0.61, *p* < 0.01), CET scores (r = 0.39; *p* < 0.01), ANSOCQ (r = −0.68, *p* < 0.01), and psychological distress (r = 0.68; *p* < 0.01). A similar significant association was observed between T1 EDE-Q global score with other T1 measures and T2 EDE-Q global score with other T2 measures (Table 7). As expected, there was a significant positive relationship between EDE-Q global score with its underlying four subscales (weight concern, dietary restrain, eating concern, and shape) at all three assessment periods (all Ps < 0.01). This indicates that greater reductions in EDE-Q subscales will eventually lead to lower overall eating disorder psychopathology.

Findings in Table 7 also found that SF12MCS score at T0 was significantly associated with participant’s age (r = 0.26; *p* < 0.05), employment status (r = 0.33, *p* < 0.01), psychotropic medication (r = −0.35; *p* < 0.01), T0 EDE-Q global score (r = −0.49; *p* < 0.01), CET scores (r = −0.26; *p* < 0.05), ANSOCQ (r = 0.33, *p* < 0.01), and psychological distress (r = −0.74; *p* < 0.01). A similar significant association was observed between T1 SF12MCS and participant’s employment status, psychotropic medication, T1 EDE-Q global score, ANSOCQ and psychological distress. T2 SF12MCS was found to be significantly associated with T2 EDE-Q global score and psychological distress (Table 7).

The EDQoL score at T0 was found to be significantly associated with T0 EDE-Q global score (r = 0.62; *p* < 0.01), CET scores (r = 0.35; *p* < 0.01), ANSOCQ (r = −0.39, *p* < 0.01), and psychological distress (r = 0.74; *p* < 0.01) (Table 7). A similar significant association was observed between T1 EDQoL and T1 EDE-Q global score, CET scores, ANSOCQ, and psychological distress. T2 EDQoL score was found to be significantly correlated with T2 EDE-Q global score, CET scores, ANSOCQ, and psychological distress. There were significant negative relationships between EDQoL and SF-12MCS at all three assessment periods (values of r ranges from −0.53 to −0.65; all Ps < 0.01), which indicates that a higher score on the SF12MCS leads to lower scores in EDQoL, i.e., a better HRQoL (Table 7).

#### 3.2.2. Correlation among Outcome Measures

The correlation matrix presented in Table 8 shows that in each period of assessment (T0, T1, T2), measures including weight concern, dietary restraint, eating concern, shape concern, CET, exercise beliefs questionnaire, ANSOCQ, and psychological distress are correlated with the respective assessment of outcome measures. At T0, the largest correlation was observed between weight concern and shape concern (r = 0.88; *p* < 0.01), followed by between ANSOCQ and shape concern (r = −0.67; *p* < 0.01), between weight concern and eating concern (r = 0.66; *p* < 0.01), and between shape concern and dietary restraint r = 0.66; *p* < 0.01). A similar correlation was observed at T1 and T2 (Table 8). All the baseline measures, including BMI, EDE-Q global score, EDQoL, SF12MCS, CET, K-10, ANSOCQ, weight concern, shape concern, and eating concern, were found to be significantly positively associated with same measures at the end of therapy and at 6-month follow-up, (Table 8). The significant correlations between and across assessment periods shows a pattern of persistence or recurrence of these psychiatric measures over the study period.

### 3.3. Results from Multiple Linear Regression Analysis

#### 3.3.1. BMI

Consistent with correlation analysis, the findings from multiple regression models (Table 9) revealed that none of the baseline (T0) measures were found to be significantly associated with BMI at T0. The regression model for the end of therapy (T1) measures revealed that after adjusting for age and treatment group, greater ANSOCQ score (β = 0.36, *p* < 0.05) and a lower value of EDQoL (β = −0.35, *p* < 0.05) were significantly associated with higher BMI at T1. The overall model for T1 was significant, accounting for 36.8% of the total variance. The regression model for the 6-month follow-up (T2) measures also shows that lower value of EDQoL (β = −0.51, *p* < 0.01) was significantly associated with a higher BMI at T2. The overall model for T2 was significant, accounting for 22.9% of the total variance (Table 9).

#### 3.3.2. EDE-Q Global Score

The regression model for the T0 measures presented in Table 9 revealed that after adjusting for age and treatment group, a lower ANSOCQ score (β = −0.50, *p* < 0.01) and a higher value of EDQoL (β = 0.29, *p* < 0.05) were significantly associated with higher EDE-Q global score at T0. The overall model at T0 for EDE-Q global score was significant, accounting for 68.6% of the total variance. The regression model for the T1 measures shows that a lower ANSOCQ score (β = −0.48, *p* < 0.01) was associated with a higher EDE-Q global score at T1. The overall model was significant, accounting for 66.2% of the total variance. The regression model for the T2 measures also found that a lower ANSOCQ score (β = −0.37, *p* < 0.01) and a higher value of EDQoL (β = 0.45, *p* < 0.01) were significantly associated with a higher EDE-Q global score at T2. The overall model was significant, accounting for 64.1% of the total variance (Table 9).

#### 3.3.3. Quality of Life

SF12MCS.: All three regression models for T0, T1, and T2 measures revealed that lower scores of psychological distress (T0: β = −0.70, *p* < 0.01; T1: β = −0.71, *p* < 0.01; T2: β = −0.81, *p* < 0.01) were significantly associated with higher SF12MCS values (Table 9). The models were significant, accounting for 52.9% of the total variance at T0; 76.4% of the total variance at T1 and 68.6% of the total variance at T2.

EDQoL: The regression model for the T0, T1, and T2 measures shows that a higher psychological distress score (T0: β = 0.61, *p* < 0.01; T1: β = 0.41, *p* < 0.01; T2: β = 0.20, *p* < 0.05) and a greater value of the EDE-Q global score (T0: β = 0.37, *p* < 0.01; T1: β = 0.34, *p* < 0.05; T2: β = 0.54, *p* < 0.01) were associated with a higher value of EDQoL (Table 9). Findings from the model for T2 measures revealed that lower value of BMI (β = −0.37, *p* < 0.01) was associated with higher EDQoL values. All three models were significant, with accounting for 59.7% of the total variance at T0; 58.4% at T1; and 71.3% of the total variance at T2.

## 4. Discussion

This paper reports in-depth secondary analysis of the first and to date only RCT in adult women to examine the efficacy of LEAP in a direct comparison with a CBT-AN condition through three successive assessments at baseline, end of therapy, and 6-month post-therapy follow-ups respectively. The findings revealed that there were large effect sizes (Cohen’s *d* >= 0.80) for EDE-Q global score, EDQoL, SF12MCS, weight concern, dietary restraint, eating concern, AN stage change, and psychological distress; medium effect sizes (Cohen’s *d* ranges 0.50 to 0.79) for shape concern; and compulsive exercise for changes between baseline to end of therapy and baseline to 6-month follow-up. The effect sizes for all measures between end of therapy and 6-month follow-up were quite small. This may have been due to the short duration of follow-up or that a small subset of participants might have experienced symptomatic worsening post-treatment. Changes between baseline and end of therapy showed that effect sizes for those in the LEAP group were slightly larger than for those in the CBT-AN group for most of the measures, including EDE-Q global score, EDQoL, weight concern, dietary restraint, and eating concern, whilst marginally larger effect sizes for CBT-AN were observed for ANSOCQ and psychological distress. The effect of therapy was observed in both LEAP and CBT-AN groups even before the 10 weeks of therapy; the EDE-Q global score, EDQoL, and SF12MCS showed significant changes between baseline and 10 weeks of therapy.

All the baseline measures, including BMI, EDE-Q global score, EDQoL, SF12MCS, CET, psychological distress, ANSOCQ, weight concern, shape concern, and eating concern were found to be significantly positively correlated with the same measures at the end of therapy and at 6-month follow-up. These correlations between and across assessment periods indicates a pattern of relative persistence or recurrence of these psychiatric measures over the study period. Both correlation analysis and the adjusted models found that a higher BMI at the end of therapy and at 6-month follow-up was significantly associated with improved EDQoL at the end of therapy and at 6-month follow-up, respectively. ANSOCQ at the end of therapy and at 6-month follow-up was also found to be significantly correlated with improved EDQoL at the end of therapy and at 6-month follow-up, respectively. The models also found that lower EDE-Q global score at all three time points was significantly associated with higher ANSOCQ and also with improved EDQoL at all three time points. Lower MHRQoL was also associated with more psychological distress at all three time points.

The present study has implications for the treatment for AN. The findings support the body of literature that, in contrast to other interventions such as medications, endorse psychological therapies for AN, showing continued improvements in symptoms over time, albeit with smaller effect sizes than during the active phase of therapy [5]. Furthermore, the results indicate that several baseline characteristics, BMI, level of ED symptoms (EDE-Q global score) and MHRQoL predict ED outcome measures, and this is consistent with the previous literature [25]. To our knowledge, the additional finding that these characteristics at end of treatment also predicted 6-month outcomes is novel. Overall, the findings suggest that assessment of these features is important not only for identifying whether a treatment is likely to be effective but also that these effects will be sustained over time. While not being predictive, two measures—psychological distress and AN stage of motivation to change—were found to be significantly associated with ED symptoms and EDQoL; again, this is consistent with the broader eating disorders research [25]. Future treatment programs should aim to reduce psychological distress and increase motivation during CBT-AN treatment because although it may not directly predict change in psychopathology at the end of therapy, participants will be in a better place for eventual recovery.

A significant limitation of this study is that there was a possible sampling bias in the recruitment from ED clinics and through community advertisements in three cities of three different countries. Due to the small number of participants overall and particularly from each country (UK: 38, Australia: 23, USA: 13), all the analyses were carried out on a combined sample which may not be representative of people receiving treatment for anorexia nervosa for each country or other countries. As with RCTs, for which there are strict inclusion criteria, results also may not be generalizable to all who have a need for treatment as this would include people with, e.g., a diagnosis of bipolar disorder or current substance dependence who were excluded from the present RCT. These study features limit the generalizability regarding the findings. Furthermore, the majority of data used for statistical analyses in this study were obtained using self-report measures. It is likely that more accurate clinical data could have been obtained by using semi-structured interviews in which the interviewer can explore symptoms experienced by participants in depth [18]. Another potential limitation of this study is that during the follow-up period, 35.1% (26 of 74) participants were lost to follow-up between baseline assessment and end of therapy assessment; overall, 44.6% (33 of 74) were lost to follow-up between baseline and 6-month follow-up assessment. Participants who were lost to follow-up between baseline and 6-month assessment periods were comparatively younger, but not significantly so, than those retained in the sample (mean age at baseline: 26.2 years vs. 27.7 years, respectively; *p* = 0.46). The differences in their reported levels of education (*p* = 0.67), employment status (*p* = 0.55), and psychotropic medication use (*p* = 0.98) at baseline were not found to be statistically significant between the two groups (Appendix D). A similar non-significant difference was also observed in baseline outcome measures, including mean body mass index (*p* = 0.84), SF12MCS (*p* = 0.15), EDQoL (*p* = 0.10), weight concern (*p* = 0.05), dietary restraint (*p* = 0.17), eating concern (*p* = 0.18), compulsive exercise test (*p* = 0.87), and exercise beliefs questionnaire (*p* = 0.37 (Appendix D). This finding indicates that the loss of 44.6% of the subjects at 6-month follow-up can assumed to have occurred at random, and the attrition is unlikely to have impacted the overall findings of this study.

Another limitation of the present study was that the level of compulsive exercise accepted for inclusion in the study was very broad, as the eligibility criteria included participants who had participated in at least one form of physical activity in the past month. Although this allows results to be generalised to a wider group of individuals with AN who engage in exercise, further investigation is needed to explore the relationship between LEAP, treatment outcomes, and their predictors for people with high levels of compulsive exercise. Furthermore, the follow-up period was only 6-months, which is relatively shorter compared to other RCT studies. Finally, it must be acknowledged that this RCT was not designed specifically for predictor or moderator analysis.

Several key strengths separate this study from previous research. This is one of the few RCTs to examine outcome measures in an outpatient adult sample and the first to examine these in a LEAP trial. The RCT followed a strict double-blind protocol. In particular, the participants were not told which treatment group they were in, therapists were trained and conducted both treatments, and adherence to protocols was assessed by reviewing audio recording of treatment sessions. The study was conducted at multiple sites on three different continents, which improves the generalisability of results. As well as this, the study explored a range of robust and well-established outcomes, including both AN-specific and general measures.

As this study was exploratory in nature, future studies should extend and replicate the findings. Future studies using larger samples could examine the predictors and other maintaining factors, specifically affect regulation, which may influence treatment outcome, as lower levels of baseline emotional dysregulation have previously predicted weight increase [38].

## 5. Conclusions

This study found large effective sizes for CBT-AN (with or without LEAP) from baseline to (a) end of therapy and (b) to 6-month follow-up in improved ED- and MH-related quality of life (MHRQoL), reduced ED symptomatology and psychological distress, and improved participant motivation to change. Medium effect sizes were found for reduced shape concern and compulsive exercise. Several baseline characteristics, including BMI, level of ED symptoms, and MHRQoL, were found to be important predictors in identifying whether a treatment is likely to be effective. Future treatment programs should aim to reduce psychological distress and increase motivation prior to CBT-AN treatment because although it may not predict change in psychopathology at the end of therapy, participants will be in a better place for eventual recovery.

## Figures and Tables

**Table 1 behavsci-13-00651-t001:** Participants’ characteristics (at baseline) by randomized group (LEAP, CBT-AN).

	Randomized Group			Significant Difference LEAP vs. CBT-AN: *p*-Values
Participant Characteristics	LEAP	CBT-AN	All Participants	
	Number (%)	Number (%)	Number (%)	
All	36 (100.0)	38 (100.0)	74 (100.0)	
Age Group				
<20 years	9 (25.0)	8 (21.1)	17 (23.0)	*p* = 0.689
20–29	19 (52.8)	18 (47.4)	37 (50.0)	*p* = 0.4638
30–39	6 (16.7)	9 (23.7)	15 (20.3)	*p* = 0.453
40 and above	2 (5.6)	3 (7.9)	5 (6.8)	*p* = 0.689
Mean Age (SD)	26.3 (8.0)	27.8 (9.5)	27.1 (8.7)	*p* = 0.478
Level of Education				
Up to High School	16 (44.4)	15 (39.5)	31 (41.9)	*p* = 0.667
Trade Certificate	3 (8.3)	3 (7.9)	6 (8.1)	*p* = 0.944
University Education	17 (47.2)	20 (52.6)	37 (50.0)	*p* = 0.638
Employment status				
Employed (part time or full time)	16 (44.4)	26 (68.4)	42 (56.6)	*p* = 0.037
Unemployed	6 (16.7)	4 (10.5)	10 (13.5)	*p* = 0.443
Student/Others	14 (38.9)	8 (21.1)	22 (29.7)	*p* = 0.092
Marital status				
Married/living as married	8 (22.2)	13 (34.2)	21 (28.4)	*p* = 0.254
Single	27 (75.0)	24 (63.2)	51 (68.9)	*p* = 0.271
Separated/divorced	1 (2.8)	1 (2.6)	2 (2.7)	*p* = 0.518
Purging Behaviour				
Yes	23 (63.9)	17 (44.7)	40 (54.1)	*p* = 0.098
No	13 (36.1)	21 (55.3)	34 (45.9)	*p* = 0.098
Psychotropic Medication				
Yes	21 (58.3)	19 (50.0)	40 (54.1)	*p* = 0.472
No	15 (41.7)	19 (50.0)	34 (45.9)	*p* = 0.472

Note: LEAP, Loughborough eating disorders activity programme; CBT-AN, cognitive behavioural therapy—anorexia nervosa.

**Table 2 behavsci-13-00651-t002:** Results of mixed models by treatment group (LEAP and CBT-AN), assessment times (three assessments: baseline, end of therapy, 6-month follow-up), and interaction of treatment group with assessment times (assessment times X group) for each outcome measures.

	Effect					
Outcome Measures	Treatment Group		Time of Assessment		Time of Assessment X Treatment Group	
	F-Value	*p*-Value	F-Value	*p*-Value	F-Value	*p*-Value
Body mass index	2.5	0.120	7.8	<0.001	2.2	0.118
EDE-Q global score	0.4	0.535	51.8	<0.001	2.0	0.139
Weight concern	1.4	0.239	37.1	<0.001	1.0	0.356
Dietary restraint	1.6	0.208	43.8	<0.001	2.3	0.101
Eating concern	0.1	0.762	33.3	<0.001	0.4	0.647
Shape concern	0.2	0.626	17.2	<0.001	2.3	0.106
Compulsive exercise test	5.0	0.028	19.9	<0.001	1.5	0.236
Exercise beliefs questionnaire	0.8	0.380	9.6	<0.001	1.1	0.351
ANSOCQ	0.6	0.456	39.7	<0.001	0.3	0.770
SF12MCS	1.9	0.173	22.1	<0.001	0.9	0.399
SF12PCS	0.9	0.356	6.8	0.002	0.2	0.851
EDQoL	0.1	0.732	35.5	<0.001	0.0	0.953
Psychological distress (K-10)	1.5	0.231	30.8	<0.001	0.7	0.521

EDE-Q global score: eating disorder examination questionnaire global score; ANSCOQ, anorexia nervosa stages of change questionnaire; SF12MCS, short-form health survey mental health subscale; SF12PCS, short-form health survey physical health; EDQoL, eating disorder quality of life. F is a statistical test statistic which examines the significant differences of mean scores.

**Table 3 behavsci-13-00651-t003:** Mean total score with 95% confidence interval (95% CI) for all outcome measures from non-matched samples for those who completed assessments at baseline (T0), end of therapy (T1), and 6-month follow-up (T2) for LEAP, CBT-AN, and all participants.

Outcome Measures: LEAP and CBT-AN Participants		Baseline (T0) Assessment		End of Therapy (T1) Assessment		6-Month Follow-Up (T2) Assessment	
		No.	Mean Score (95% CI)	No.	Mean Score (95% CI)	No.	Mean Score (95% CI)
Body mass index							
LEAP		36	16.6 (16.2, 16.9)	22	16.7 (14.8, 18.6)	16	18.0 (17.0, 18.9) *↑^b^*
CBT-AN		38	16.5 (16.1, 16.9)	26	18.3 (17.4, 19.2) *↑^b^*	23	18.3 (17.2, 19.4) *↑^b^*
All participants *^#^*		74	16.5 (16.3, 16.8)	48	17.6 (16.6, 18.6)	39	18.2 (17.5, 18.9) ↑^b^
LEAP vs. CBT-AN *p*-values from *t*-test			0.913		0.107		0.658
EDE-Q global score							
LEAP		36	4.1 (3.7, 4.6)	22	2.2 (1.6, 2.7) ↓^a^	17	2.5 (1.7, 3.3) ↓^a^
CBT-AN		38	4.0 (3.5, 4.4)	25	2.5 (1.8, 3.2) ↓^a^	24	2.1 (1.5, 2.7) ↓^a^
All participants		74	4.1 (3.7, 4.4)	47	2.3 (1.9, 2.8) ↓^a^	41	2.3 (1.8, 2.7) *↓^a^*
LEAP vs. CBT-AN: *p*-values from *t*-test			0.632		0.476		0.384
Weight concern							
LEAP		36	4.0 (3.5, 4.5)	20	2.3 (1.5, 3.0) ↓^a^	12	2.0 (1.1, 2.8) ↓^a^
CBT-AN		38	3.3 (2.8, 3.8)	24	2.2 (1.5, 3.0)	23	1.9 (1.2, 2.5) ↓^a^
All participants		74	3.6 (3.3, 4.0)	44	2.2 (1.7, 2.7) ↓^a^	35	1.9 (1.4, 2.4) *↓^a^*
LEAP vs. CBT-AN: *p*-values from *t*-test			*0.068*		*0.930*		*0.834*
**Dietary restraint**							
LEAP		36	3.8 (3.3, 4.2)	20	1.5 (0.8, 2.2) ↓^a^	12	1.8 (1.0, 2.6) ↓^a^
CBT-AN		38	3.8 (3.4, 4.2)	24	2.6 (2.0, 3.2) ↓^a^	23	1.9 (1.3, 2.6) ↓^a^
All participants		74	3.8 (3.5, 4.1)	44	2.1 (1.6, 2.5) *↓^a^*	35	1.9 (1.4, 2.4) *↓^a^*
LEAP vs. CBT-AN: *p*-values from *t*-test			0.818		0.020		0.772
Eating concern							
LEAP		36	2.9 (2.4, 3.3)	20	1.3 (0.6, 1.9) ↓^a^	12	0.8 (0.4, 1.2) ↓^a^
CBT-AN		38	2.7 (2.3, 3.2)	24	1.7 (1.1, 2.3)	23	1.3 (0.6, 1.9) ↓^a^
All participants		74	2.8 (2.5, 3.1)	44	1.5 (1.1, 1.9) *↓^a^*	35	1.1 (0.7, 1.6) *↓^a^*
LEAP vs. CBT-AN: *p*-values from *t*-test			0.654		0.351		0.337
**Shape concern**							
LEAP		36	3.7 (3.2, 4.3)	20	2.2 (1.5, 2.9) ↓^a^	12	1.9 (1.2, 2.7) ↓^a^
CBT-AN		38	3.1 (2.6, 3.6)	24	2.5 (1.8, 2.2) ↓^a^	23	2.1 (1.3, 2.9) ↓^a^
All participants		74	3.4 (3.0, 3.8)	44	2.4 (1.9, 2.9) *↓^a^*	35	2.1 (1.5, 2.6) *↓^a^*
LEAP vs. CBT-AN: *p*-values from *t*-test			0.085		0.612		0.750
**Compulsive exercise test**							
LEAP	36		16.6 (16.2, 16.9)	20	10.3 (8.5, 12.2) ↓^a^	15	9.9 (7.5, 12.3)
CBT-AN	38		16.5 (16.1, 16.9)	25	14.5 (12.3, 16.6)	24	14.3 (11.8, 16.7)
All participants ^#^	74		16.5 (16.3, 16.8)	45	12.6 (11.1, 14.2) ↓^a^	39	12.6 (10.7, 14.4) ↓^a^
LEAP vs. CBT-AN: *p*-values from *t*-test			*0.335*		*0.006*		*0.016*
Exercise beliefs questionnaire							
LEAP	35		44.7 (37.3, 52.1)	20	25.2 (15.4, 35.1)	14	28.7 (16.3, 41.0)
CBT-AN	38		45.6 (37.5, 53.8)	25	40.0 (28.6, 51.3)	24	44.0 (32.2, 55.8)
All participants	73		45.2 (39.8, 50.6)	45	33.4 (25.7, 41.1)	38	38.4 (29.7, 47.1)
LEAP vs. CBT-AN: *p*-values from *t*-test			0.865		0.054		0.084
ANSOCQ							
LEAP	36		2.4 (2.1, 2.6)	20	3.4 (3.0, 3.9) *↑^b^*	15	3.3 (2.6, 3.9)
CBT-AN	38		2.4 (2.2, 2.6)	25	3.5 (3.1, 3.9) *↑^b^*	24	3.4 (3.0, 3.8)
All participants	74		2.4 (2.2, 2.5)	45	3.5 (3.2, 3.7) *↑^b^*	39	3.4 (3.0, 3.7)
LEAP vs. CBT-AN: *p*-values from *t*-test			0.742		0.838		0.660
SF12MCS							
LEAP	33		27.8 (23.8, 31.7)	20	37.1 (32.7. 41.5) *↑^b^*	15	38.7 (31.5, 46.0)
CBT-AN	37		29.0 (24.9, 33.0)	24	40.4 (35.1, 45.6) *↑^b^*	24	41.0 (35.5, 46.5) *↑^b^*
All participants	70		28.4 (25.6, 31.2)	44	38.9 (35.5, 42.3) *↑^b^*	39	40.1 (35.9, 44.3) *↑^b^*
LEAP vs. CBT-AN: *p*-values from *t*-test			0.676		0.343		0.607
SF12PCS							
LEAP	33		47.5 (44.1, 50.8)	20	52.8 (49.3. 56.4)	15	52.3 (47.9, 56.7)
CBT-AN	37		46.4 (42.7, 50.1)	24	51.0 (47.3, 54.8)	24	50.1 (45.4, 54.8)
**All participants**	70		46.9 (44.5, 49.4)	44	51.8 (49.3, 54.4)	39	51.0 (47.7, 54.2)
LEAP vs. CBT-AN: *p*-values from *t*-test			0.651		0.484		0.513
EDQoL							
LEAP	36		1.7 (1.5, 1.9)	20	0.9 (0.6, 1.2) ↓^a^	15	0.8 (0.4, 1.2) ↓^a^
CBT-AN	38		1.8 (1.5, 2.0)	25	1.0 (0.7, 1.3) ↓^a^	24	0.9 (0.6, 1.2) ↓^a^
**All participants**	74		1.7 (1.6, 1.9)	45	1.0 (0.7, 1.2) ↓^a^	39	0.9 (0.6, 1.1) ↓^a^
LEAP vs. CBT-AN: *p*-values from *t*-test			0.647		0.648		0.689
Psychological distress (K10)							
LEAP	36		31.8 (28.7, 35.0)	20	23.2 (18.8, 27.5) ↓^a^	15	21.5 (16.5, 26.5) ↓^a^
CBT-AN	38		30.7 (27.5, 33.8)	25	21.6 (17.7, 25.5) ↓^a^	24	21.6 (18.0, 25.1) ↓^a^
Total	74		31.2 (29.1, 33.4)	45	22.3 (19.5, 25.1) ↓^a^	39	21.6 (18.8, 24.3) ↓^a^
LEAP vs. CBT-AN: *p*-values from *t*-test			0.601		0.582		0.992

Note: LEAP, Loughborough eating disorders activity programme; CBT-AN, cognitive behavioural therapy—anorexia nervosa. EDE-Q global score: eating disorder examination questionnaire global Score; K10, the Kessler-10 Psychological Distress Scale; ANSCOQ, anorexia nervosa stages of change questionnaire; SF12MCS, short-form health survey mental health subscale; EDQoL, eating disorder quality of life. Multiple group comparison test (Bonferroni test) between T0 vs. T1, T0 vs. T2, and T1 vs. T2 were conducted through ANOVA for repeated measures (T1, T2, T3); the t-test was used to examine the significant differences of mean scores between the LEAP and CBT-AN groups. ↓^a^ Indicates that mean score is significantly (*p* < 0.05) lower than baseline (T0) score. *↑**^b^*** Indicates that mean score is significantly (*p* < 0.05) higher than baseline (T0) score. **^#^** Due to exclusion of missing or not stated cases, the total number of participants at baseline, at the end of therapy, and at 6-month follow-up may slightly differ in outcome measures.

**Table 4 behavsci-13-00651-t004:** Effect size estimates (Cohen’s *d* with 95% CI) for treatment outcomes between baseline (T0) and end of therapy (T1); between T0 and 6-month follow-up (T2) assessment; and between T1 and T2 matched samples for LEAP, CBT-AN, and all participants, respectively.

	Effect Size: Cohens’ *d* (95%CI)		
Outcome Measures:	LEAP Participants	CBT-AN Participants	All Participants
Body mass index			
Between baseline and end of therapy assessment	0.04 (−0.38, 0.46)	0.73 (0.29, 1.16)	0.29 (0.01, 0.58)
Between baseline and 6-month follow-up assessment	0.87 (0.28, 1.44)	0.66 (0.20, 1.11)	0.71 (0.36, 1.06)
Between end of therapy and 6-month follow-up assessment	0.23 (−0.28, 0.74)	0.26 (−0.17, 0.68)	0.20 (−0.13, 0.53)
EDE-Q global score			
Between baseline and end of therapy assessment	−1.55 (−2.17, −0.92)	−1.02 (−1.50, 0.53)	−1.24 (−1.62, −0.85)
Between baseline and 6-month follow-up assessment	−0.75 (−1.28, −0.20)	−1.11 (−1.62, 0.60)	−0.96 (−1.33, 0.58)
Between end of therapy and 6-month follow-up assessment	0.38 (−0.13, 0.88)	−0.35 (−0.79, 0.08)	−0.09 (−0.41, 0.22)
EDQoL			
Between baseline and end of therapy assessment	−1.21 (−1.79, −0.63)	−0.88 (−1.33, −0.41)	−1.00 (−1.35, −0.64)
Between baseline and 6-month follow-up assessment	−0.86 (−1.45, −0.26)	−0.84 (−1.30, 0.36)	−0.86 (−1.22, −0.48)
Between end of therapy and 6-month follow-up assessment	−0.09 (−0.58, 0.41)	−0.28 (−0.70, 0.15)	−0.21 (−0.54, 0.12)
SF12MCS			
Between baseline and end of therapy assessment	0.71 (0.20, 1.21)	1.09 (0.56, 1.60)	0.91 (0.54, 1.26)
Between baseline and 6-month follow-up assessment	0.33 (−0.21, 0.86)	0.87 (0.38, 1.34)	0.63 (0.27, 0.97)
Between end of therapy and 6-month follow-up assessment	−0.01 (−0.50, 0.49)	0.19 (−0.24, 0.62)	0.11 (−0.22, 0.44).
Weight concern			
Between baseline and end of therapy assessment	−1.08 (−1.62, −0.51)	−0.80 (−1.40, −0.32)	−0.92 (−1.27, −0.56)
Between baseline and 6-month follow-up assessment	−1.02 (−1,02, −0,30)	−0.96 (−14.5, −0.46)	−0.99 (−1.39, −0.58)
Between end of therapy and 6-month follow-up assessment	−0.10 (−0.67, 0.47)	−0.38 (−0.80, 0.06)	−0.27 (−0.61, 0.07)
Dietary restraint			
Between baseline and end of therapy assessment	−1.14 (−1.70, −0.56)	−0.94 (−1.42,,−0.45)	−1.03 (−1.39, −0.66)
Between baseline and 6-month follow-up assessment	−0.81 (−1.45, −0.14)	−1.02 (−1.52, −0.51)	−0.96 (−1.35, −0.55)
Between end of therapy and 6-month follow-up assessment	−0.05 (−0.61, 0.52)	−0.49 (−0.93, −0.04)	−0.31 (−0.65, 0.03)
Eating concern			
Between baseline and end of therapy assessment	−0.84 (−1.35, −0.32)	−0.82 (−1.27, −0.32)	−0.84 (−1.18, −0.49)
Between baseline and 6-month follow-up assessment	−0.83 (−1.49, −0.16)	−0.92 (−0.140, −0.42)	−0.90 (−1.29, −0.51)
Between end of therapy and 6-month follow-up assessment	−0.39 (−0.97, 0.21)	−0.41 (−0.84, 0.03)	−0.40 (−0.75, −0.05)
Shape concern			
Between baseline and end of therapy assessment	−0.80 (−1.30, −0.29)	−0.36 (−0.77, 0.06)	−0.57 (−0.88, −0.24)
Between baseline and 6-month follow-up assessment	−0.83 (−1.48, −0.16)	−0.47 (−0.90, −0.04)	−0.60 (−0.95, −0.23)
Between end of therapy and 6-month follow-up assessment	−0.16 (−0.73, 0.41)	−0.32 (−0.74, 0.11)	−0.27 (−0.61, 0.07)
Compulsive exercise test			
Between baseline and end of therapy assessment	−0.98 (−1.50, −0.43)	−0.58 (−1.00, −0.15)	−0.74 (−1.07, −0.41)
Between baseline and 6-month follow-up assessment	−0.72 (−1.28, −0.14)	−0.60 (−1.03, −0.16)	−0.65 (−1.00, −0.30)
Between end of therapy and 6-month follow-up assessment	0.11 (−0.42, 0.63)	−0.06 (−0.48, 0.35)	−0.02 (−0.35, 0.30)
Exercise beliefs questionnaire			
Between baseline and end of therapy assessment	−0.77 (−1.27, −0.26)	−0.40 (−0.80, 0.02)	−0.54 (−0.85, −0.22)
Between baseline and 6-month follow-up assessment	−0.40 (−0.94, 0.15)	−0.34 (−0.75, 0.08)	−0.37 (−0.69, −0.04)
Between end of therapy and 6-month follow-up assessment	0.36 (−0.20, 0.92)	0.26 (−0.17, 0.68)	0.30 (−0.04, 0.63)
ANSOCQ			
Between baseline and end of therapy assessment	1.01 (0.46, 1.55)	1.05 (0.55, 1.53)	1.04 (0.68, 1.40)
Between baseline and 6-month follow-up assessment	0.63 (0.06, 1.11)	0.90 (0.41, 1.37)	0.80 (0.43, 1.15)
Between end of therapy and 6-month follow-up assessment	−0.27 (−0.80, 0.27)	−0.05 (0.47, 0.37)	−0.13 (−0.46, 0.19)
Psychological distress (K10)			
Between baseline and end of therapy assessment	−0.83 (−1.33, 0.31)	−1.07 (−1.56, −0.57)	−0.96 (−1.31, −0.60)
Between baseline and 6-month follow-up assessment	−0.60 (−1.09, 01)	−0.92 (−1.40, 0.44)	−0.77 (−1.13, −0.41)
Between end of therapy and 6-month follow-up assessment	0.23 (−0.28, 0.72)	−0.16 (−0.59, 0.26)	0.03 (−0.30, 0.35)

**Note:** LEAP, Loughborough eating disorders activity program; CBT-AN, cognitive behavioural therapy—anorexia nervosa; EDEQ global score: eating disorder examination questionnaire global score. Cohen’s *d*: The effect size (Cohen’s *d*) for individual measures were calculated by comparing the baseline (T0) to end of therapy (T1); baseline (T0) with 6-month follow-up (T2); end of therapy (T1) with 6-month follow-up (T2). Cohen’s *d* indicates small effect = 0.20; medium effect = 0.50; large effect = 0.80.

**Table 5 behavsci-13-00651-t005:** Effect size estimates (Cohens’ *d*) with 95% confidence interval (95% CI) for four main treatment outcomes between baseline (T0) and 6-month follow-up (T2) assessment for all participants by socio-demographic characteristics by LEAP and CBT-AN.

	Body Mass Index		EDE-Q Global Score		SF12-MCS		EDQoL	
	LEAP	CBT-AN	LEAP	CBT-AN	LEAP	CBT-AN	LEAP	CBT-AN
	Cohens’ *d* (95%CI)	Cohens’ *d* (95%CI)	Cohens’ *d* (95%CI)	Cohens’ *d* (95%CI)	Cohens’ *d* (95%CI)	Cohens’ *d* (95%CI)	Cohens’ *d* (95%CI)	Cohens’ *d* (95%CI)
Age group								
<20 years	1.28 (−0.39, 2.85)	0.81 (0.07, 1.51)	−1.12 (−2.38, 0.22)	−0.37 (−1.00, 0.28)	1.10 (−0.46, 2.56)	−0.04 (−0.74, 0.65)	−1.71 (−3.60, 0.23)	−0.47 (−1.15, 0.23)
20–29	0.24 (−0.94, 1.37)	0.90 (0.21, 1.57)	−0.66 (−1.89, 0.67)	−1.49 (−2.31, −0.64)	1.17 (−0.86, 3.07)	1.40 (0.58, 2.19)	−0.50 (−1.67, 0.77)	−1.14 (−1.86, −0.39)
30 and above	0.53 (−0.75, 1.71)	0.47 (−0.28, 1.19)	−2.98 (−5.90, −0.13)	−0.96 (−1.74, −0.14)	0.44 (−0.81, 1.59)	0.42 (−028, 1.09)	−2.39 (−4.82, 0.03)	−0.81 (−1.55, −0.03)
Level of Education								
Up to High School	1.02 (−0,02, 2.00)	0.76 (0.04, 1.46)	−0.84 (−1.76, 0.13)	−0.66 (1.30, 0.01)	0.50 (−0.47, 1.41)	0.19 (−0.47, 0.85)	−1.01 (−1.98, 0.03)	−0.74 (−1.47, 0.02)
University Education	0.59 (−0.18, 1.33)	0.72 (0.14, 1.28)	−1.01 (−1.81, −0.18)	−1.16 (−1.81, −0.49)	1.05 (0.15, 1.90)	0.76 (0.17, 1.32)	−0.76 (−1.49, 0.01)	−0.87 (−1.46, −0.26)
Employment status								
Employed	0.39 (−0.34, 1.10)	1.62 (0.52, 2.68)	−1.11 (−1.98, −0.19)	−0.51 (−1.19, 0.21)	0.10 (−0.65, 0.85)	0.51 (−0.30, 1.28)	−1.19 (−2.09, −0.24)	−0.68 (−1.49, 0.17)
Unemployed/other	0.66 (0.11, 1.20)	0.70 (−0.16, 1.52)	−1.13 (−1.73, −0.50)	−1.04 (−1.95, −0.008)	0.72 (0.18, 1.25)	1.34 (0.17, 2.45)	−0.93 (−1.49, −0.34)	−0.90 (−1.76, 0.02)
Purging Behaviour								
Yes	1.03 (01.9, 1.82)	0.67 (−0.18, 1.48)	−0.65 (1.36, −1.36)	−0.83 (−1.62, 0.01)	0.26 (0.42, 0.91)	0.43 (−0.51, 1.34)	−0.76 (−1.49, 0.01)	−0.95 (−1.90, 0.06)
No	0.98 (0.15, 1.77)	0.50 (−0.07, 1.05)	−1.68 (−2.65, −0.68)	−0.88 (−1.49, −0.25)	1.15 (0.27, 1.99)	0.71 (0.11, 1.29)	−1.10 (−1.88, −0.28)	−0.72 (−1.30, −0.12)
Psychotropic Medication								
Yes	0.98 (0,20, 1.73)	0.65 (−0.27, 1.52)	−0.58 (−1.21, 0.08)	−1.07 (−2.06, −0.01)	0.25 (−0.47, 0.94)	0.59 (−0.31, 1.44)	−0.70 (−1.72, 0.05)	−1.23 (−2.28, −0.11)
No	0.99 (0.16, 1.78)	0.60 (0.02, 1.16)	−1.62 (−2.56, −0.64)	−0.91 (−1.52, −0.27)	0.84 (0.10, 1.56)	0.85 (0.20, 1.48)	−0.68 (−1.36, 0.03)	−0.93 (−1.55, −0.29)

**Table 6 behavsci-13-00651-t006:** Mean total score with standard deviation (SD) for main eating disorder examination (EDE) outcome measures (body mass index, EDE-Q global score, SF12MCS, EDQoL) at baseline by patient’s characteristics.

Patient Characteristics	No. of Cases	Body Mass Index	EDE-Q Global Score	SF12MCS	EDQoL
		Mean (SD)	Mean (SD)	Mean (SD)	Mean (SD)
All participants	74	16.5 (1.1)	4.1 (1.4)	28.4 (11.6)	1.7 (0.7)
Randomised group					
LEAP	36	16.6 (1.1)	4.1 (1.3)	27.8 (11.2)	1.7 (0.6)
CBT-AN	38	16.5 (1.2)	4.0 (1.4)	28.9 (12.2)	1.8 (0.8)
*t*-test: *p*-values		0.913	0.6323	0.676	0.6478
Age group					
<20 years	17	16.4 (1.0)	4.5 (1.5)	24.4 (10.0)	1.9 (0.7)
20–29	37	16.6 (1.2)	3.9 (1.2)	28.3 (11.8)	1.6 (0.6)
30–39	15	16.2 (1.1)	4.1 (1.3)	32.4 (13.0)	1.9 (0.8)
40 and above	5	17.2 (0.8)	3.3 (1.9)	30.8 (10.0)	1.3 (0.6)
*F*-test: *p*-values		0.300	0.313	0.291	0.144
Level of education					
Up to high school	31	16.5 (1.1)	4.4 (1.2)	25.0 (9.6)	1.8 (0.6)
Trade certificate	6	16.9 (0.5)	3.8 (1.3)	30.8 (9.9)	1.7 (0.4)
University education	37	16.5 (1.2)	3.8 (1.5)	30.7 (12.9)	1.7 (0.8)
*F*-test: *p*-values		0.749	0.256	0.136	0.885
Employment status					
Employed	42	16.5 (1.1)	3.9 (1.4)	31.8 (11.6)	1.6 (0.6)
Unemployed/others	32	16.5 (1.1)	4.3 (1.2)	23.9 (10.2)	1.9 (0.8)
*t*-test: *p*-values		0.992	0.178	0.004	0.054
Purging behaviour					
Yes	40	16.5 (1.1)	4.6 (1.0)	28.5 (11.7)	1.8 (0.7)
No	34	16.5 (1.1)	3.4 (1.4)	28.3 (11.7)	1.6 (0.7)
*t*-test: *p*-values		0.859	0.001	0.941	0.181
Psychotropic medication					
Yes	40	16.5 (1.2)	4.3 (1.4)	25.2 (10.3)	1.8 (0.7)
No	34	16.6 (1.1)	3.8 (1.3)	32.5 (12.1)	1.6 (0.7)
*t*-test: *p*-values		0.920	0.116	0.008	0.088

Note: SD, Standard deviation; K10, the Kessler-10 Psychological Distress Scale; SF12MCS, short-form health survey mental health subscale; EDQoL, eating disorder quality of life. *F* and *t*-tests were conducted to examine the significant differences of mean scores across patient characteristics.

**Table 7 behavsci-13-00651-t007:** Correlations between main outcome measures themselves (body mass index, EDE-Q global score, SF12MCS, EDQoL) and correlation (associations) between selected measures or predictors with main outcome measures as well as and between main outcome measures at baseline (T0), end of therapy (T1), and 6-months follow-up (T2) assessments.

Assessment and Measures	Baseline Assessment (n = 74)			
Baseline Assessment	Body Mass Index	EDE-Q Global Score	SF12MCS	EDQoL
Body mass index	-	0.10	−0.01	−0.02
EDE-Q global score	0.10	-	−0.49 **	0.62 **
*SF12MCS*	−0.01	−0.49 **	-	−0.53 ****
EDQoL	−0.02	0.61 **	−0.53 ****	-
Age (in years)	0.04	−0.26 *	0.33 **	−0.09
Employed (no = 0, yes = 1)	−0.04	−0.19	0.35 **	−0.21
Purging behaviour (no = 0, yes = 1)	0.02	0.47 **	0.01	0.16
Psychotropic medication (no = 0, yes = 1)	0.001	0.22	−0.35 **	0.222
Weight concern	0.04	0.83 **	−0.44 **	0.55 **
Dietary restraint	0.06	0.68 **	0.46 **	0.38 **
Eating concern	−0.08	0.63 **	−0.42 **	0.49 **
Shape concern	0.11	0.79 **	−0.39 **	0.52 **
Compulsive exercise test	0.12	0.39 **	−0.26 *	0.35 **
Exercise beliefs questionnaire	0.12	0.19	−0.05	0.19
ANSOCQ	−0.02	−0.68 **	0.33 **	−0.39 **
Psychological distress (K10)	−0.08	0.68 **	−0.74 **	0.74 **
	End of therapy assessment (n = 48)			
End of therapy assessment	Body mass index	EDE-Q global score	SF12MCS	EDQoL
Body mass index	-	−0.15	0.30 *	−0.48 **
EDE-Q global score	−0.15	-	−0.66 **	0.65 **
SF12MCS	0.30 *	−0.66 **	-	−0.65 **
EDQoL	−0.48 **	0.65 **	−0.65 **	-
Age (in years)	−0.06	−0.02	−0.01	0.16
Employed (no = 0, yes = 1)	0.04	−0.23	0.30 *	−0.07
Purging behaviour (no = 0, yes = 1)	0.25	0.30 *	−0.19	0.04
Psychotropic medication (no = 0, yes = 1)	−0.26	0.18	−0.32 *	0.20
Weight concern	−0.12	*0.75 ***	−0.71 **	0.55 **
Dietary restraint	−0.15	*0.60 ***	−0.42 **	0.45 **
Eating concern	−0.29	*0.68 ***	−0.53 **	0.57 **
Shape concern	−0.09	*0.79 ***	−0.60 **	0.55 **
Compulsive exercise test	−0.20	0.56 **	−0.23	0.46 **
Exercise beliefs questionnaire	−0.22	0.32 *	−0.06	0.41 **
ANSOCQ	0.34 *	−0.72 **	0.41 **	−0.44 **
Psychological distress (K10)	−0.29	0.58 **	−0.86 **	0.68 **
	6-Month follow-up assessment (n = 41)			
6-month follow-up assessment	Body mass index	EDE-Q global score	SF12MCS	EDQoL
Body mass index	-	−0.21	0.20	−0.52 **
EDE-Q global score	−0.21	-	−0.53 **	0.74 **
SF12MCS	0.20	−0.53 **		−0.57 **
EDQoL	−0.52 **	0.74 **	−0.57 **	-
Age (in years)	−0.15	−0.04	−0.07	0.02
Employed (no = 0, yes = 1)	−0.06	−0.29	0.24	−0.11
Purging behaviour (no = 0, yes = 1)	0.17	0.23	0.02	0.02
Psychotropic medication (no = 0, yes = 1)	−0.08	0.30	−0.30	0.18
Weight concern	−0.23	0.70 **	−0.46 **	0.65 **
Dietary restraint	−0.39 *	0.77 **	−0.39 *	0.67 **
Eating concern	−0.45 *	0.72 **	−0.48 **	0.81 **
Shape concern	−0.09	0.70 **	−0.39 *	0.55 **
Compulsive exercise test (CET)	−0.24	0.43 **	−0.17	0.53 **
Exercise beliefs questionnaire	−0.21	0.21	−0.02	0.36 *
ANSOCQ	0.35 *	−0.64 **	0.26	−0.46 **
Psychological distress (K-10)	−0.12	0.60 **	−0.84 **	0.58 **

Note: EDE-Q global: eating disorder examination questionnaire global score; SF12MCS, short-form health survey mental health subscale; EDQoL, eating disorder quality of life; K-10, the Kessler-10 Psychological Distress Scale. ***** Indicates significant at *p* < 0.05; ****** indicates significant at *p* < 0.01.

**Table 8 behavsci-13-00651-t008:** Correlations between selected outcome measures (or predictor measures) themselves at baseline (T0), end of therapy (T1), and 6-monthsfollow-up (T2) assessments.

Assessment and Measures	Baseline Assessment							
Baseline Assessment	Weight Concern	Dietary Restraint	Eating Concern	Shape Concern	Compulsive Exercise Test	Exercise Beliefs	Stage of Change	K10
Weight concern	1.00							
Dietary restraint	0.65 **	1.00						
Eating concern	0.66 **	0.64 **	1.00					
Shape concern	0.88 **	0.66 **	0.60 **	1.00				
Compulsive exercise test	0.34 **	0.29 **	0.37 **	0.31 **	1.00			
Exercise beliefs questionnaire	0.14	0.10	0.24 *	0.15	0.61 **	1.00		
ANSOCQ	−0.62 **	−0.54 **	−0.44 **	−0.67 **	−0.33 **	−0.13	1.00	
Psychological distress (K-10)	0.63 **	0.59 **	0.64 **	0.62 **	0.29 *	0.14	−0.49 **	1.00
	End of therapy assessment							
End of therapy assessment	Weight concern	Dietary restraint	Eating concern	Shape concern	Compulsive exercise test	Exercise beliefs	Stage of change	K-10
Weight concern	1.00							
Dietary restraint	0.53 **	1.00						
Eating concern	0.77 **	0.56 **	1.00					
Shape concern	0.83 **	0.57 **	0.66 **	1.00				
Compulsive exercise test	0.41 *	0.49 **	0.49 **	0.54 **	1.00			
Exercise beliefs questionnaire	0.23	0.45 **	0.32 *	0.28	0.73 **	1.00		
Stage of change	−0.53 **	−0.52 **	−0.36 *	−0.70 **	−0.54 **	−0.34 *	1.00	
Psychological distress (K10)	0.69 **	0.36 *	0.48 **	0.54 **	0.25	0.17	−0.36 *	1.00
	6-month follow-up Assessment							
6-month follow-up assessment	Weight concern	Dietary restraint	Eating concern	Shape concern	Compulsive exercise test	Exercise beliefs	Stage of change	K-10
Weight concern	1.00							
Dietary restraint	0.62 **	1.00						
Eating concern	0.72 **	0.73 **	1.00					
Shape concern	0.82 **	0.52 **	0.57 **	1.00				
Compulsive exercise test	0.38 *	0.42 *	0.47 **	0.39 *	1.00			
Exercise beliefs questionnaire	0.26	0.32	0.35	0.35	0.78 **	1.00		
ANSOCQ	−0.55 **	−0.54 **	−0.46 **	−0.47 **	−0.44 **	−0.36 *	1.00	
Psychological distress (K-10)	0.64 **	0.58 **	0.64 **	0.58 **	0.23	0.18	−0.35 *	1.00

* Indicates significant at *p* < 0.05; ** indicates significant at *p* < 0.01.

**Table 9 behavsci-13-00651-t009:** Multiple linear regression analysis: standardised regression coefficients (beta, *β*) for predictors of body mass index, EDE-Q global score, SF12MCS, and EDQoL at baseline (T0), end of therapy (T1), and 6-month follow-up (T2) assessments.

Assessment and Measures ##	Baseline Assessment (n = 74): Standardised Regression Coefficients (*β*)			
Baseline Assessment	Body Mass Index	EDE-Q Global Score	SF12MCS	EDQoL
	** *β* **	** *β* **	** *β* **	** *β* **
Treatment group (0 = LEAP, 1 = CBT-AN)	0.24	−0.06	−0.04	0.10
Age (in years)	−0.02	−0.04	0.11	0.05
Employed (no = 0, yes = 1)		-	0.12	
Psychotropic medication (no = 0, yes = 1)		-	−0.04	
Compulsive exercise test		0.16 *	−0.17	0.06
Exercise beliefs questionnaire				
ANSOCQ		−0.50 **	−0.01	0.20
Psychological distress (K10)		0.11	−0.70 **	0.61 **
Body mass index	-	-		−0.02
EDE-Q global score		-	0.14	0.37 **
SF12MCS			-	
EDQoL		0.29 **		-
Adjusted R square	0.014	0.686 **	0.529 **	0.597 **
	End of therapy assessment (n = 48)			
End of therapy assessment	Body mass index	EDE-Q global score	SF12MCS	EDQoL
	** *β* **	** *β* **	** *β* **	** *β* **
Treatment group (0 = LEAP, 1 = CBT-AN)	0.21	0.12	0.10	0.04
Age (in years)	−0.10	−0.09	0.09	0.05
Employed (no = 0, yes = 1)			−0.14	
Psychotropic medication (no = 0, yes = 1)			−0.09	
Compulsive exercise test		0.21		0.02
Exercise beliefs questionnaire		−0.15		0.17
ANSOCQ	0.36 *	−0.48 **	−0.09	0.133
Psychological distress (K-10)		0.23	−0.71 **	0.41 **
Body mass index	-		0.05	−0.24
EDE-Q global score			−0.34 *	0.34 *
SF12MCS			-	
EDQoL	−0.35 *	0.25	NI	-
Adjusted R square	0.368 **	0.662 **	0.764 **	0.584 **
	6-month follow-up assessment (n = 41)			
6-month follow-up assessment	Body mass index	EDE-Q global score	SF12MCS	EDQoL
	** *β* **	** *β* **	** *β* **	** *β* **
Treatment group (0 = LEAP, 1 = CBT-AN)	0.16	−0.05	0.09	0.07
Age (in years)	−0.12	−0.08	−0.01	−0.06
Employed (no = 0, yes = 1)				
Psychotropic medication (no = 0, yes = 1)				
Compulsive exercise test		0.01		0.23
Exercise beliefs questionnaire				0.01
ANSOCQ	0.05	−0.37 **		0.14
Psychological distress (K-10)		0.21	−0.81 **	0.20
Body mass index	-			−0.37 **
EDE-Q global score			−0.05	0.54 **
SF12MCS				
EDQoL	−0.51 **	0.45 **		-
Adjusted R square	0.229 *	0.641 **	0.686 **	0.713 **

Note: EDE-Q global score: eating disorder examination questionnaire global score; SF12MCS, short-form health survey mental health subscale; EDQoL, eating disorder quality of life; K-10, the Kessler-10 Psychological Distress Scale. **##** Statistically significant predictors/measures found in bivariate and correlation analysis (Table 6 and Table 7) are included in multiple linear regression analysis. All the models were adjusted for the randomisation group and participants’ age. ***** Indicates significant at *p* < 0.05; ****** indicates significant at *p* < 0.01.

## Data Availability

The data presented in this study are available on request from the corresponding author. The data are not publicly available due to privacy reasons.

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
